# Association between Prior Cytotoxic Therapy, Antecedent Hematologic Disorder, and Outcome after Allogeneic Hematopoietic Cell Transplantation in Adult Acute Myeloid Leukemia

**DOI:** 10.3390/cancers15020352

**Published:** 2023-01-05

**Authors:** Corentin Orvain, Eduardo Rodríguez-Arbolí, Megan Othus, Brenda M. Sandmaier, H. Joachim Deeg, Frederick R. Appelbaum, Roland B. Walter

**Affiliations:** 1Clinical Research Division, Fred Hutchinson Cancer Center, Seattle, WA 98109, USA; 2Department of Medicine, Division of Hematology, University of Washington, Seattle, WA 98195, USA; 3Maladies du Sang, CHU d’Angers, 49000 Angers, France; 4Fédération Hospitalo-Universitaire Grand-Ouest Acute Leukemia, FHU-GOAL, 49033 Angers, France; 5Inserm UMR 1307, CNRS UMR 6075, Université d’Angers, CRCI2NA, Nantes Université, 49000 Angers, France; 6Department of Hematology, Hospital Universitario Virgen del Rocío, Instituto de Biomedicina de Sevilla (IBIS/CSIC/CIBERONC), University of Seville, 41013 Seville, Spain; 7Public Health Sciences Division, Fred Hutchinson Cancer Center, Seattle, WA 98109, USA; 8Department of Medicine, Division of Medical Oncology, University of Washington, Seattle, WA 98109, USA; 9Department of Laboratory Medicine & Pathology, University of Washington, Seattle, WA 98195, USA; 10Department of Epidemiology, University of Washington, Seattle, WA 98195, USA

**Keywords:** acute myeloid leukemia (AML), allogeneic hematopoietic cell transplantation (HCT), therapy-related, antecedent hematologic disease (AHD), prognostication

## Abstract

**Simple Summary:**

To assess the association between clinical history and AML outcomes in the context of allogeneic hematopoietic cell transplantation (HCT), we retrospectively analyzed 739 patients with de novo AML, 125 with antecedent hematologic disorder (AHD)/AML, and 115 with therapy-related AML who received first allografts while in first or second remission. Relative to patients with de novo AML, relapse rates were similar for patients with AHD and therapy-related AML after multivariable adjustment, as were relapse-free survival and overall survival. Non-relapse mortality was, however, higher for AHD AML. These data suggest that the clinical history by itself contains limited prognostic value for adults with AML undergoing allografting, supporting the most recent approach to use this information as a diagnostic qualifier rather than a disease entity.

**Abstract:**

(1) Background: Secondary acute myeloid leukemia (AML), i.e., AML arising from prior therapy (therapy-related) and/or an antecedent hematologic disorder (AHD) is generally associated with worse outcomes compared to de novo AML. However, recognizing the prognostic importance of genetic characteristics rather than clinical history, secondary AML is now considered a diagnostic qualifier rather than a separate disease entity. (2) Methods: To assess the association between clinical history and AML outcomes in the context of allogeneic hematopoietic cell transplantation (HCT), we retrospectively analyzed 759 patients with de novo AML, 115 with AHD AML, and 105 with therapy-related AML who received first allografts while in first or second remission. (3) Results: At the time of HCT, these three cohorts differed significantly regarding many patient and disease-specific characteristics, including age (*p* < 0.001), gender (*p* < 0.001), disease risk (*p* = 0.005), HCT-CI score (*p* < 0.001), blood count recovery (*p* = 0.003), first vs. second remission (*p* < 0.001), remission duration (*p* < 0.001), measurable residual disease (MRD; *p* < 0.001), and conditioning intensity (*p* < 0.001). Relative to patients with de novo AML, relapse rates were similar for patients with AHD (hazard ratio [HR] = 1.07, *p* = 0.7) and therapy-related AML (HR = 0.86, *p* = 0.4) after multivariable adjustment, as were relapse-free survival (HR = 1.20, *p* = 0.2, and HR = 0.89, *p* = 0.5) and overall survival (HR = 1.19, *p* = 0.2, and HR = 0.93, *p* = 0.6). Non-relapse mortality was higher for AHD AML (HR = 1.59, *p* = 0.047). (4) Conclusions: These data suggest that the clinical history by itself contains limited prognostic value for adults with AML undergoing allografting, supporting the most recent approach to use this information as a diagnostic qualifier rather than a disease entity.

## 1. Introduction

Secondary acute myeloid leukemia (AML), i.e., AML arising after prior exposure to cytotoxic therapy (therapy-related AML) and/or antecedent hematological disorder (AHD), has generally been associated with a worse prognosis compared to de novo AML [1,2,3,4,5,6]. However, it is increasingly appreciated that genetics and patient characteristics rather than clinical history largely account for outcome differences [1,2,3,5,6,7,8]. As a result, the International Consensus Classification, the World Health Organization (WHO), and European LeukemiaNet (ELN) now consider secondary AML as a disease attribute and diagnostic qualifier rather than a separate disease entity [9,10,11]. For most adults with de novo or secondary AML, allogeneic hematopoietic cell transplantation (HCT) is an important part of curative-intent treatment [1,2,4,12,13,14,15]. In some [16,17,18] but not all studies [19], clinical history had been associated with post-HCT outcome, with increased relapse risk due to high-risk disease features (e.g., progressive disease, adverse cytogenetic/molecular abnormalities) and, possibly, pre-HCT measurable residual disease (MRD) in individuals with secondary AML [16,20,21]. In other studies, secondary AML was associated with higher non-relapse mortality (NRM) [17].

Given these mixed data, and the notion that post-HCT outcomes within the subset of patients with secondary AML might differ depending on the details of clinical history (e.g., whether therapy-related or after AHD, or the type of underlying AHD) [17,22], we examined a large cohort of adults with therapy-related AML and after AHD who underwent allogeneic HCT while in first or second remission at our institution between May 2006 and May 2021. In this analysis, we also assessed how the most recent changes in the classification for secondary AML might impact study findings.

## 2. Materials and Methods

### 2.1. Study Cohort

We included all adults with AML (2016 WHO criteria) [23] who proceeded to first allogeneic HCT while in first or second remission (i.e., had <5% marrow blasts) with or without peripheral blood count recovery between 5/2006 and 5/2021 at a single institution (Fred Hutchinson Cancer Center/University of Washington/Seattle Cancer Care Alliance, Seattle, WA, USA). High-dose fractionated total body irradiation (TBI; ≥12 Gy) with or without cyclophosphamide (CY) or fludarabine (FLU), high-dose TBI/thiotepa/FLU, busulfan (4 days) with CY or FLU, treosulfan/FLU with or without low-dose TBI, or any regimen containing a radiolabeled antibody were considered myeloablative conditioning (MAC) regimens [24]. Reduced-intensity conditioning (RIC) and non-myeloablative conditioning (i.e., 2–3 Gy TBI with or without fludarabine) were grouped together as non-MAC regimens. Data on post-HCT outcomes were obtained via the Long-Term Follow-Up Program from our outpatient clinic and local clinics that provided care for post-HCT patients; additionally, information was collected on patients in research studies. All treatments were on Institutional Review Board-approved research protocols (all registered within ClinicalTrials.gov) or standard protocols; patients gave informed consent in accordance with the Declaration of Helsinki. The cut-off date for follow-up was 10 February 2022.

### 2.2. Classification of Disease Risk and Treatment Response

Secondary AML was defined as disease following an AHD (i.e., myelodysplastic syndrome [MDS], myeloproliferative neoplasm [NPM], and MDS/MPN such as chronic myelomonocytic leukemia [CMML]) or treatment with systemic chemotherapy and/or radiotherapy for a different disorder. All previous cytotoxic regimens were considered for the main analysis, including methotrexate, mercaptopurine, and cyclophosphamide for auto-immune disease. Immunosuppressive treatment using nonchemotherapeutic agents was not considered cytotoxic. Patients developing MDS, MPN, or MDS/MPN between the chemotherapy or radiation treatment for their primary disease and the diagnosis of AML were classified as therapy-related AML unless stated otherwise. In sensitivity analyses, we explored the impact of new classification criteria for therapy-related AML (not including previous exposure to methotrexate, and, by analogy, mercaptopurine, and cyclophosphamide) and AHD (requirement for AHD diagnosis ≥3 months before AML diagnosis) [10,11]. The 2010 refined MRC/NCRI criteria were used to assess the cytogenetic risk at diagnosis [25]. The karyotype analysis was routinely based on 20 metaphases (if available); FISH studies were performed according to standard procedures in a subset of patients.

The HCT-specific comorbidity index (HCT-CI) and the Treatment-Related Mortality (TRM) score were calculated as previously described [26,27]. We categorized treatment responses according to the ELN (2017) except that we defined post-HCT relapse as emergence >5% blasts in blood or bone marrow, new cytogenetic abnormality, or any level of disease that led to a therapeutic intervention [28]. Multiparameter (10-color) flow cytometry was routinely performed on bone marrow before initiating conditioning therapy. The methodology of the MFC MRD assay and its performance has not changed throughout the study period, [29,30,31,32,33,34,35], with any measurable level of residual disease considered MRD-positive [29,30,31,32,33,35,36,37,38,39].

### 2.3. Statistical Analysis

Categorical variables were reported as numbers (with proportions) and compared using the Chi² test or Fisher’s exact test, as appropriate. Continuous variables were reported as medians (with interquartile range [IQR]) and compared using the Mann and Whitney test. We estimated unadjusted probabilities of relapse-free survival (RFS; events: relapse or death) and overall survival (OS; event: death) with the Kaplan–Meier method and compared them with the Log-Rank test. Associations with RFS and OS were assessed using Cox regression models. Probabilities of relapse (with NRM as a competing event) and NRM (death without relapse with relapse as a competing event) were reported using cumulative incidence estimates. Associations with relapse and NRM were assessed using cause-specific regression models [24]. All tests were two-sided with a significant level of *p* < 0.05. Statistical analyses were performed with R (R Foundation for Statistical Computing, Vienna, Austria; http://www.r-project.org).

## 3. Results

### 3.1. Characteristics of Study Cohort

Of 1011 adults undergoing a first allogeneic HCT for AML in first or second remission, 21 did not agree to data use for research while data from institutional pre-HCT MRD testing were not available for 11 patients. Among the remaining 979 patients, 115 patients had therapy-related AML, mainly following treatment for lymphoid hematologic malignancy (*n* = 39, 34%), breast cancer (*n* = 35, 30%), or an auto-immune condition (*n* = 13, 11%), and 125 patients had an AHD (MDS, *n* = 97, 78%; MPN, *n* = 18, 14%; CMML, *n* = 10, 8%; Appendix A). Among patients with AHD, 20 (16%) occurred post cytotoxic therapy for other conditions, and 50 (40%) were previously treated for AHD.

The characteristics of the 979 patients included in the analysis are summarized in Table 1. Patients with AHD were older than patients with either de novo or therapy-related AML (63 vs. 53 vs. 57 years, *p* < 0.001). Due to the high number of therapy-related AML after treatment for breast cancers, there were more females in the therapy-related group (60% vs. 46% vs. 35% for patients with de novo and AHD, respectively, *p* < 0.001) and, as prior cancer is used to calculate the HCT-CI, patients in this group were more likely considered as high-risk according to this score (68% vs. 25% vs. 27%, respectively, *p* < 0.001). On the other hand, patients with AHD were less likely to be classified as favorable risk (0 vs. 8% vs. 11% for patients with de novo and therapy-related AML, respectively, *p* = 0.005) and they were more likely to have incomplete blood count recovery (43% vs. 28% vs. 34%, respectively, *p* = 0.003) and pre-HCT MRD (38% vs. 16% vs. 24%, respectively, *p* < 0.001) whereas the time from the last remission to HCT was shorter (72 vs. 104 vs. 96 days, respectively, *p* < 0.001). Most patients with secondary AML, either therapy-related or AHD, were transplanted in first remission (91% and 87% vs. 73% for patients with de novo AML, *p* < 0.001) and they were more likely to receive non-MAC regimens (53% and 56% vs. 36%, respectively, *p* < 0.001).

As the definitions of sAML were recently changed [10,11], we also compared the three groups after reclassifying five patients with previous auto-immune conditions treated with either methotrexate (*n* = 3), and, by analogy, mercaptopurine (*n* = 1) or cyclophosphamide (*n* = 1), and 34 patients with AHD diagnosed ≤3 months before AML diagnosis (MDS, *n* = 28; CMML, *n* = 3; MPN, *n* = 3) as de novo AML. As summarized in Appendix A, the characteristics of the three patient subsets did not significantly change with this reclassification.

### 3.2. Relationship between Secondary AML Status and Post-HCT Outcome

After a median follow-up of 5.13 years after HCT among survivors (IQR: 2.30–9.01), there were 308 relapses, 460 deaths, and 191 NRM events. Relapse was non significantly higher in patients with AHD (35% [26–45%] vs. 23% [20–26%] vs. 28% [20–36%] at three years for those with de novo and therapy-related AML, respectively, Log-Rank test: *p* = 0.099; Figure 1) as was NRM (24% [15–32%] vs. 16% [13–19%] vs. 15% [8–22%] at three years, *p* = 0.058). This translated into statistically significant lower RFS (39% [31–50%] vs. 55% [52–59%] vs. 50% [42–61%] at three years, *p* < 0.001) and OS (43% [34–54%] vs. 61% [58–65%] vs. 58% [49–68%] at three years, *p* = 0.001) in patients with AHD. Using the updated classification schemes for secondary AML moderately modified these observations, with relapse being statistically significantly higher in patients with AHD (44% [32–56%] vs. 29% [26–32%] vs. 34% [25–43%] at three years, *p* = 0.044; Appendix A). As the 34 patients not considered as having AHD in this last analysis were more likely to have prior MDS or not have been previously treated for the AHD, we explored whether these characteristics were associated with post-HCT outcomes. After considering all patients with AHD, including those with prior exposure to cytotoxic therapy, neither type of AHD nor previous therapy for AHD were associated with post-HCT outcome (Appendix A).

### 3.3. AML after AHD as an Independent Prognostic Factor for Post-HCT Outcome

To study the relationship between AML after AHD and post-HCT outcomes in more detail, we evaluated univariable and multivariable regression models for the endpoints of NRM, relapse, RFS, and OS. In univariable analysis, AML after AHD was associated with NRM (hazard ratio [HR] = 1.94 [1.30–2.88], *p* = 0.001), relapse (HR = 1.54 [1.11–2.15], *p* = 0.011), RFS (HR = 1.69 [1.31–2.18], *p* < 0.001), and OS (HR = 1.63 [1.25–2.12], *p* < 0.001) whereas therapy-related AML was not (Table 2). Similar results were observed when limiting patients with AHD to those with a three-month interval between AHD and AML diagnosis (as performed in the 2022 classification scheme), including NRM (HR = 2.13 [1.35–3.37], *p* = 0.001), relapse (HR = 1.81 [1.24–2.63], *p* = 0.002), RFS (HR = 1.93 [1.44–2.57], *p* < 0.001), and OS (HR = 1.69 [1.24–2.29], *p* < 0.001). Similar C-statistic values (indicating similar predictive accuracy) were observed when considering AHD without any time requirement before AML diagnosis or with a three-month time requirement (0.53 vs. 0.53 for NRM, 0.53 vs. 0.53 for relapse, 0.53 vs. 0.53 for RFS, and 0.53 vs. 0.52 for OS).

**Figure 1 cancers-15-00352-f001:**
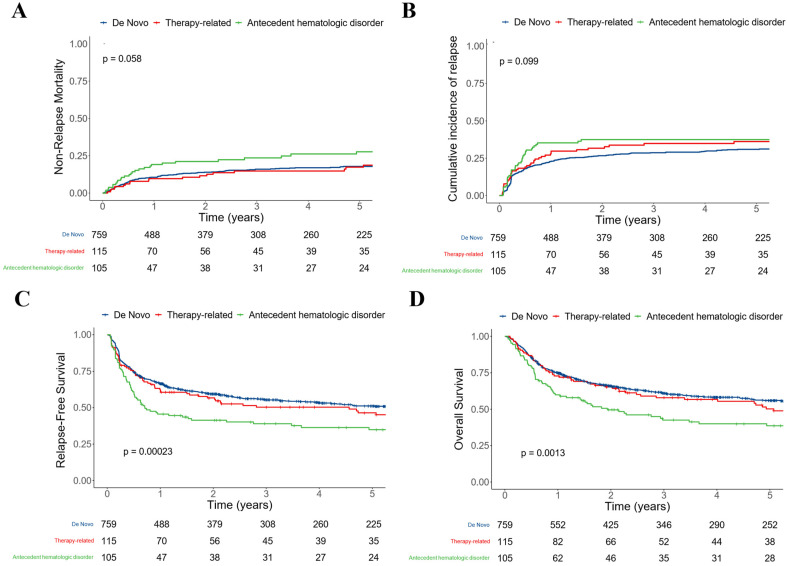
Post-HCT outcomes for 979 adults with AML undergoing allogeneic HCT while in first or second morphologic remission, stratified by disease status at diagnosis (de novo vs. therapy-related vs. antecedent hematologic disorder); (**A**) Non-relapse mortality; (**B**) relapse; (**C**) relapse-free survival; and (**D**) overall survival.

After multivariable adjustment, AML after AHD remained significantly associated with NRM (HR = 1.59 [1.01–2.52], *p* = 0.047) but not relapse (HR = 1.07 [0.73–1.58], *p* = 0.7), RFS (HR = 1.20 [0.89–1.61], *p* = 0.2), or OS (HR = 1.19 [0.88–1.62], *p* = 0.2; Table 3). As AML after AHD was not independently associated with post-HCT outcomes other than NRM, we sought to identify variables that were associated with the outcome by performing univariable regression models in patients with AHD (*n* = 126). In these patients, adverse cytogenetic risk (HR = 1.80 [1.01–3.20], *p* = 0.046) and positive pre-HCT MRD by MFC (HR = 2.21 [1.28–3.81], *p* = 0.005) were associated with relapse and positive pre-HCT MRD was associated with RFS (HR = 1.70 [1.10–2.61], *p* = 0.016) and OS (HR = 1.75 [1.12–2.72], *p* = 0.014; Table 4). As patients with AHD were significantly older, we specifically explored the association between age and post-HCT outcomes, dichotomizing patients into those younger vs. older than 60 years old. While the NRM for younger and older patients was similar (at three years: 22% [10–34%] vs. 19% [7–31%], *p* = 0.99), as was OS (at three years: 55% [42–71%] vs. 38% [29–52%], *p* = 0.38), relapse was non-significantly higher (at three years: 26% [14–39%] vs. 48% [37–60%], *p* = 0.08) and RFS was non-significantly lower (at three years: 52% [39–68%] vs. 33% [24–45%], *p* = 0.07) in younger patients. In contrast to the whole study cohort, non-MAC was not associated with NRM, RFS, or OS.

### 3.4. Relationship between Disease Status, Conditioning Intensity, and Post-HCT Outcomes

To further explore the relationship between disease status and post-HCT outcomes, we performed subset analyses in which we studied patients who underwent transplants after MAC or non-MAC conditioning separately. These analyses were further motivated by a significant interaction between conditioning intensity (MAC vs. non-MAC) and AHD for NRM (*p* = 0.043) but not relapse (*p* = 0.2), RFS (*p* = 0.8), or OS (*p* = 0.085). In patients who received MAC, AHD was significantly associated with NRM (26% [14–39%] vs. 11% [8–14%] vs. 11% [2–20%] at three years for those with de novo and therapy-related AML, respectively, *p* = 0.006) but not for relapse (27% [14–39%] vs. 28% [24–32%] vs. 31% [17–44%] at three years, respectively, *p* = 0.997; Figure 2). This translated into significantly lower RFS (47% [34–63%] vs. 61% [57–66%] vs. 59% [46–75%] at three years, respectively, *p* = 0.039) and OS (46% [34–63%] vs. 66% [62–70%] vs. 64% [52–80%] at three years, respectively, *p* = 0.0029). After multivariable adjustment, AHD remained statistically associated with NRM (HR = 2.71 [1.38–5.34], *p* = 0.004) but not with relapse (HR = 0.58 [0.31–1.07], *p* = 0.081), RFS (HR = 0.96 [0.61–1.52], *p* = 0.9), or OS (HR = 1.34 [0.84–2.13], *p* = 0.2) in patients receiving MAC. On the other hand, in patients who received non-MAC, AHD was not significantly associated with NRM (21% [10–32%] vs. 25% [19–30%] vs. 18% [8–28%] at three years, respectively, *p* = 0.628) but was associated with relapse (47% [34–61%] vs. 30% [25–36%] vs. 39% [26–51%] at three years, respectively, *p* = 0.038; Figure 3). This translated into non-significantly lower RFS (32% [22–48%] vs. 45% [39–52%] vs. 44% [33–58%] at three years, respectively, *p* = 0.064) whereas OS was similar (40% [28–56%] vs. 53% [47–59%] vs. 53% [42–67%] at three years, respectively, *p* = 0.63) in the subset of patients with AHD AML. Similar results were observed when distinguishing RIC from non-myeloablative conditioning (Appendix A). After multivariable adjustment, AHD remained statistically associated with relapse (HR = 1.86 [1.11–3.13], *p* = 0.018) and showed a borderline association with RFS (HR = 1.47 [0.98–2.22], *p* = 0.062), but not NRM (HR = 1.19 [0.62–2.27], *p* = 0.6) or OS (HR = 1.15 [0.75–1.75], *p* = 0.5).

## 4. Discussion

Recently introduced classification systems consider secondary AML as a diagnostic qualifier, in line with the increasing understanding that underlying genetic characteristics rather than clinical history largely account for worse outcomes seen in this patient subset [9,10,11]. Consistent with this notion, in our large retrospective analyses, we found very little evidence of an independent prognostic role for clinical history in multivariable analyses, except for a significant interaction between conditioning intensity and AML arising from AHD. Specifically, we found that compared to patients with de novo or therapy-related AML, those with AML arising from an AHD had a significantly higher risk of NRM if treated with MAC (but this did not translate into differences in post-transplant relapse rates or OS).

In the 2022 disease classifications (Internal Consensus Classification, WHO), some of the parameters for when exactly an AML should be considered secondary have changed. Specifically, it is now proposed that patients who developed AML after receiving cytotoxic therapy such as methotrexate for auto-immune conditions, or patients for whom the time interval between AHD and AML diagnosis was less than three months, should no longer be considered as secondary AML. Nevertheless, for our main analysis, we chose to consider these patients as secondary AML for comparability with prior studies [1,2,4,12,16,17,18,19,22,40,41]. Complicating comparability, some of the older studies already used the 3-month time interval [5,15], but none classified AML cases after prior exposure to cytotoxic therapy such as methotrexate as de novo. However, we then performed sensitivity analyses considering these classification changes, which revealed a limited impact on the characteristics of each patient subgroup and the observed post-HCT outcomes. In these sensitivity analyses, our main findings were largely unchanged. As multilineage dysplasia is no longer recognized as a prognostic feature [42], these patients were considered as having de novo AML unless they were previously diagnosed with an AHD or had prior exposure to cytotoxic therapy.

Without adjustments, we observed that AML after AHD but not therapy-related AML was associated with post-HCT outcomes in our cohort. This finding is consistent with a recent report on a cohort of patients developing myeloid neoplasms after prior therapy for solid tumors [43]. The finding contrasts, however, with one previous report showing that both patients with AHD and therapy-related AML had inferior outcomes compared to patients with de novo AML [17]. A large part of these outcome differences could be attributed to differences in patient/disease characteristics. Patients with AHD were, on average, older and less likely to have favorable-risk cytogenetics but more likely to have pre-HCT MRD by MFC than the other patients. These differences explained why AHD was no longer associated with relapse or RFS after multivariable analyses [16,19], although one study previously observed an independent association between secondary AML and relapse [17].

As in previous reports [6,16,17], we observed in our study that secondary AML, and more specifically, AHD, was independently associated with NRM, suggesting that patients with AHD experienced more complications after transplant. Although there is no clear explanation for this observation with potential patient differences that are not fully captured in our multivariable adjustments, it is supported by the fact that the association between AHD and NRM was mostly observed in patients receiving MAC. As patients previously treated for AHD had similar outcomes to those who were not treated, increased exposure to pre-HCT therapies does not appear to have a role in increased NRM. Despite being older, patients with AHD had a similar number of comorbidities, as assessed by the HCT-CI. This may represent a bias toward selecting fitter subsets of patients for allografting. As stated, the association between AHD and NRM was attenuated after multivariable adjustment for other variables associated with increased NRM such as age.

Of note, conditioning intensity was not equally associated with post-HCT outcomes in patients with AHD. As in one previous report [16], but not another from the EBMT registry [44], we observed that patients with secondary AML receiving MAC were more likely to have increased NRM even after multivariable adjustment. In contrast to that previous report, but similar to a study from the EBMT registry [44], patients who received non-MAC had increased relapse and decreased RFS in our cohort, albeit not statistically significantly so after multivariable adjustment. Although we usually recommend that patients who are fit to tolerate MAC should receive a high-intensity conditioning regimen, more caution should be applied in patients with AHD. The intensity of conditioning was not associated with post-HCT outcomes in patients with therapy-related AML, which was observed in one study [45], but not in two others [46,47], in which relapse risk was higher in patients receiving non-MAC regimens.

The retrospective nature of our study analyzing post-HCT outcomes of patients for whom conditioning intensity was nonrandomly assigned limits our ability to draw definitive conclusions regarding the management of patients with AHD AML. Our general management has been to use MAC whenever it was felt that it could be safely administered based on patient age and comorbidities. Another limitation is the fact that mutational profiles were available only for a small subset of patients.

## 5. Conclusions

The fact that AML developed post cytotoxic therapy or after AHD provides little prognostic value (except for an increased risk of NRM in patients with AHD) supports the most recent proposal to use this information largely as a diagnostic qualifier rather than a prognostic variable.

## Figures and Tables

**Figure 2 cancers-15-00352-f002:**
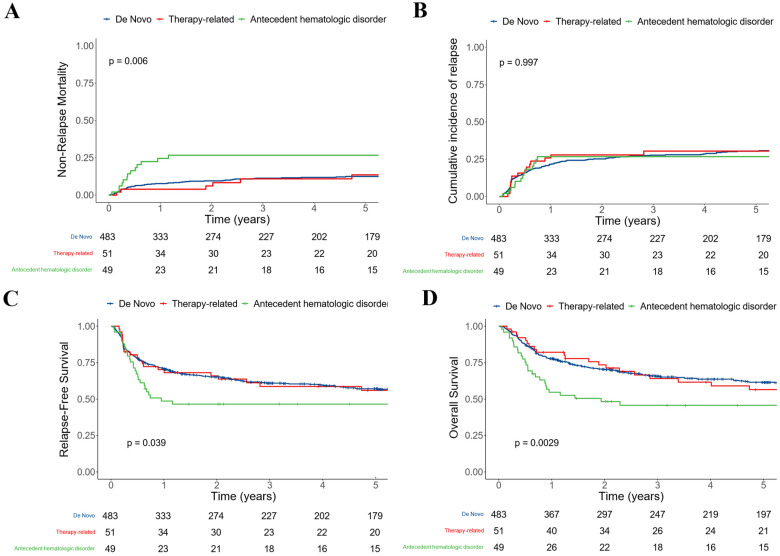
Post-HCT outcomes for 583 adults with AML undergoing allogeneic HCT following myeloablative conditioning while in first or second morphologic remission, stratified by disease status at diagnosis (de novo vs. therapy-related vs. antecedent hematologic disorder): (**A**) Non-relapse mortality; (**B**) relapse; (**C**) relapse-free survival; and (**D**) overall survival.

**Figure 3 cancers-15-00352-f003:**
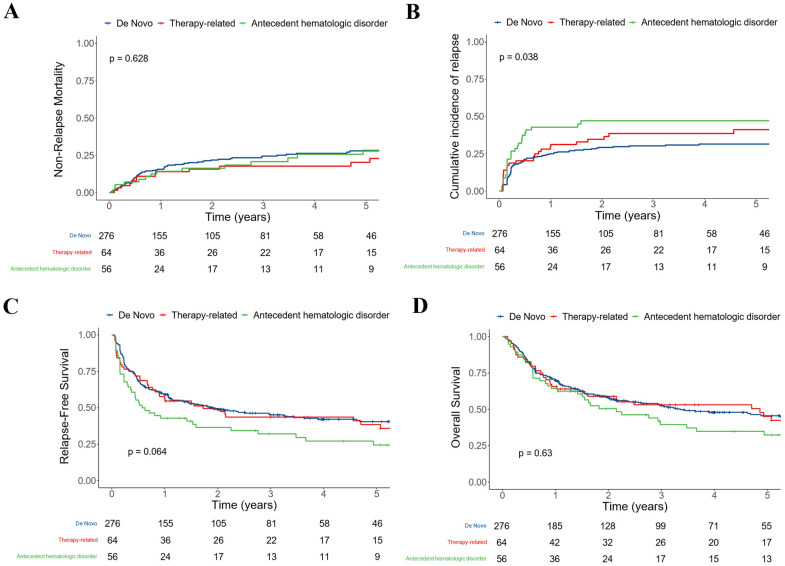
Post-HCT outcomes for 396 adults with AML undergoing allogeneic HCT following non-MAC while in first or second morphologic remission, stratified by disease status at diagnosis (de novo vs. therapy-related vs. antecedent hematologic disorder): (**A**) Non-relapse mortality; (**B**) relapse; (**C**) relapse-free survival; and (**D**) overall survival.

**Table 1 cancers-15-00352-t001:** Pre-HCT demographic and clinical characteristics of study cohort (*n* = 979), stratified according to disease status at diagnosis (de novo vs. therapy-related vs. after antecedent hematologic disorder [AHD]).

Characteristic	All Patients (*n* = 979)	De Novo (*n* = 759)	Therapy-Related (*n* = 115)	AHD (*n* = 105)
Age at HCT (IQR), years	55 (42–64)	53 (40–62)	57 (48–66)	63 (56–68)
Female gender, *n* (%)	454 (46%)	348 (46%)	69 (60%)	37 (35%)
WBC count at diagnosis (IQR), G/L	8 (2–40)	9 (2–48)	5 (2–33)	3 (2–14)
Cytogenetic risk (MRC), *n* (%)				
Favorable	72 (8%)	60 (8%)	12 (11%)	0
Intermediate	659 (70%)	520 (71%)	68 (61%)	71 (72%)
Adverse	213 (23%)	153 (21%)	32 (29%)	28 (28%)
Time from last remission to HCT (IQR), days	98 (69–146)	104 (72–150)	96 (69–132)	72 (53–105)
Disease status at HCT, *n* (%)				
First remission	747 (76%)	551 (73%)	105 (91%)	91 (87%)
Second remission	232 (24%)	208 (27%)	10 (8.7%)	14 (13%)
MFC MRD status before HCT, *n* (%)				
MRD-negative	788 (80%)	636 (84%)	87 (76%)	65 (62%)
MRD-positive	191 (20%)	123 (16%)	28 (24%)	40 (38%)
Recovered peripheral blood counts before HCT, *n* (%)	680 (69%)	544 (72%)	76 (66%)	60 (57%)
HCT-CI category, *n* (%)				
Low	339 (35%)	293 (39%)	8 (7.0%)	38 (36%)
Intermediate	347 (35%)	279 (37%)	29 (25%)	39 (37%)
High	293 (30%)	187 (25%)	78 (68%)	28 (27%)
Stem cell source, *n* (%)				
BM	81 (8%)	68 (9%)	8 (7%)	5 (5%)
PBSC	766 (78%)	584 (77%)	91 (79%)	91 (87%)
Cord blood	132 (13%)	107 (14%)	16 (14%)	9 (9%)
HLA matching, *n* (%)				
Identical related donor	227 (23%)	173 (23%)	35 (30%)	19 (18%)
Matched unrelated donor	482 (49%)	369 (49%)	56 (49%)	57 (54%)
1–2 allele mismatch	101 (10%)	79 (10%)	5 (4%)	17 (16%)
Haplo-identical	37 (4%)	31 (4%)	3 (3%)	3 (3%)
Cord blood	132 (13%)	107 (14%)	16 (14%)	9 (9%)
Conditioning regimen intensity, *n* (%)				
MAC	583 (60%)	483 (64%)	51 (44%)	49 (47%)
Non-MAC	396 (40%)	276 (36%)	64 (56%)	56 (53%)

Abbreviations: BM, bone marrow; ELN, European LeukemiaNet; HCT, hematopoietic cell transplantation; HCT-CI, HCT comorbidity index; HLA, human leukocyte antigen; MAC, myeloablative conditioning; MFC, multiparameter flow cytometry; MRC, U.K. Medical Research Council; MRD, measurable residual disease; PBSC, peripheral blood stem cells; WBC, white blood cell count.

**Table 2 cancers-15-00352-t002:** Univariable regression models of study cohort.

	Non-Relapse Mortality		Relapse		RFS		OS	
	HR (95% CI)	*p*	HR (95% CI)	*p*	HR (95% CI)	*p*	HR (95% CI)	*p*
Age at transplantation, years	1.04 (1.02–1.05)	<0.001	1.01 (1.00–1.02)	0.035	1.02 (1.01–1.03)	<0.001	1.02 (1.01–1.03)	<0.001
Female gender	0.71 (0.54–0.95)	0.022	0.85 (0.68–1.06)	0.2	0.79 (0.67–0.95)	0.011	0.81 (0.67–0.97)	0.022
WBC count at diagnosis, G/L	1.00 (1.00–1.00)	0.3	1.00 (1.00–1.00)	0.5	1.00 (1.00–1.00)	0.3	1.00 (1.00–1.00)	0.2
Disease status at diagnosis								
De novo	Ref.		Ref.		Ref.		Ref.	
Therapy-related	1.06 (0.68–1.66)	0.88	1.19 (0.85–1.67)	0.3	1.14 (0.88–1.50)	0.3	1.12 (0.84–1.48)	0.4
Antecedent hematologic disorder	1.94 (1.30–2.88)	0.001	1.54 (1.11–2.15)	0.011	1.69 (1.31–2.18)	<0.001	1.63 (1.25–2.12)	<0.001
Cytogenetic risk (MRC)								
Favorable	Ref.		Ref.		Ref.		Ref.	
Intermediate	1.43 (0.81–2.52)	0.2	1.75 (0.99–3.06)	0.052	1.59 (1.06–2.37)	0.024	1.74 (1.13–2.69)	0.012
Adverse	0.98 (0.51–1.89)	>0.9	3.20 (1.79–5.71)	<0.001	2.11 (1.38–3.21)	<0.001	2.22 (1.40–3.50)	<0.001
Time from last remission to HCT, days	1.00 (1.00–1.00)	0.5	1.00 (1.00–1.00)	0.034	1.00 (1.00–1.00)	0.2	1.00 (1.00–1.00)	0.3
Disease status at HCT								
First remission	Ref.		Ref.		Ref.		Ref.	
Second remission	1.20 (0.87–1.66)	0.3	1.36 (1.06–1.74)	0.016	1.30 (1.06–1.58)	0.010	1.34 (1.09–1.64)	0.005
MFC MRD status before HCT								
MRD-negative	Ref.		Ref.		Ref.		Ref.	
MRD-positive	1.60 (1.10–2.32)	0.014	4.28 (3.40–5.40)	<0.001	3.11 (2.57–3.76)	<0.001	2.65 (2.18–3.23)	<0.001
Recovered peripheral blood counts before HCT	0.60 (0.45–0.80)	<0.001	0.95 (0.74–1.21)	0.7	0.79 (0.66–0.95)	0.014	0.76 (0.63–0.92)	0.006
HCT-CI category								
Low	Ref.		Ref.		Ref.		Ref.	
Intermediate	1.25 (0.87–1.79)	0.2	1.10 (0.84–1.44)	0.5	1.15 (0.93–1.43)	0.2	1.17 (0.93–1.46)	0.2
High	1.70 (1.19–2.42)	0.003	1.16 (0.87–1.53)	0.3	1.34 (1.08–1.67)	0.008	1.41 (1.12–1.77)	0.003
Stem cell source								
BM	Ref.		Ref.		Ref.		Ref.	
PBSC	1.91 (0.97–3.74)	0.060	0.69 (0.48–0.98)	0.036	0.93 (0.68–1.27)	0.6	0.93 (0.67–1.27)	0.6
Cord blood	2.03 (0.96–4.30)	0.065	0.56 (0.35–0.90)	0.017	0.85 (0.58–1.25)	0.4	0.94 (0.63–1.39)	0.7
HLA matching								
Identical related donor	Ref.		Ref.		Ref.		Ref.	
Matched unrelated donor	1.09 (0.75–1.59)	0.6	0.97 (0.74–1.28)	0.8	1.02 (0.81–1.27)	0.9	1.04 (0.82–1.32)	0.7
1–2 allele mismatch	2.53 (1.61–3.96)	<0.001	1.15 (0.77–1.73)	0.5	1.61 (1.20–2.17)	0.002	1.80 (1.32–2.44)	<0.001
Haplo-identical	1.48 (0.63–3.50)	0.4	1.74 (1.02–2.94)	0.040	1.68 (1.08–2.64)	0.023	1.75 (1.08–2.82)	0.022
Cord blood	1.37 (0.85–2.22)	0.2	0.81 (0.54–1.22)	0.3	1.00 (0.73–1.36)	>0.9	1.13 (0.82–1.56)	0.4
Conditioning regimen intensity								
MAC	Ref.		Ref.		Ref.		Ref.	
Non-MAC	2.28 (1.72–3.04)	<0.001	1.36 (1.08–1.70)	0.008	1.66 (1.39–1.98)	<0.001	1.60 (1.33–1.92)	<0.001

Abbreviations: BM, bone marrow; ELN, European LeukemiaNet; HCT, hematopoietic cell transplantation; HCT-CI, HCT comorbidity index; HLA, human leukocyte antigen; MAC, myeloablative conditioning; MFC, multiparameter flow cytometry; MRC, U.K. Medical Research Council; MRD, measurable residual disease; PBSC, peripheral blood stem cells; WBC, white blood cell count.

**Table 3 cancers-15-00352-t003:** Multivariable regression models of study cohort.

	Non-Relapse Mortality		Relapse		RFS		OS	
	HR (95% CI)	*p*	HR (95% CI)	*p*	HR (95% CI)	*p*	HR (95% CI)	*p*
Age at transplantation, years	1.02 (1.00–1.04)	0.012	1.00 (0.99–1.01)	0.4	1.00 (1.00–1.01)	0.3	1.01 (1.00–1.01)	0.2
Female gender	0.75 (0.55–1.03)	0.071	1.02 (0.80–1.30)	0.9	0.91 (0.75–1.10)	0.3	0.87 (0.71–1.06)	0.2
WBC count at diagnosis, G/L	1.00 (1.00–1.01)	0.068	1.00 (1.00–1.00)	0.013	1.00 (1.00–1.00)	0.002	1.00 (1.00–1.00)	0.004
Disease status at diagnosis								
De novo	Ref.		Ref.		Ref.		Ref.	
Therapy-related	0.84 (0.51–1.39)	0.5	0.86 (0.59–1.26)	0.4	0.89 (0.66–1.21)	0.5	0.93 (0.68–1.27)	0.6
Antecedent hematologic disorder	1.59 (1.01–2.52)	0.047	1.07 (0.73–1.58)	0.7	1.20 (0.89–1.61)	0.2	1.19 (0.88–1.62)	0.2
Cytogenetic risk (MRC)								
Favorable	Ref.		Ref.		Ref.		Ref.	
Intermediate	1.28 (0.69–2.37)	0.4	1.97 (1.06–3.65)	0.031	1.58 (1.02–2.43)	0.040	1.74 (1.09–2.77)	0.020
Adverse	0.83 (0.40–1.72)	0.6	3.20 (1.66–6.17)	<0.001	1.93 (1.20–3.10)	0.007	2.07 (1.24–3.44)	0.005
Time from last remission to HCT, days	1.00 (1.00–1.00)	>0.9	1.00 (1.00–1.00)	0.5	1.00 (1.00–1.00)	0.7	1.00 (1.00–1.00)	0.7
Disease status at HCT								
First remission	Ref.		Ref.		Ref.		Ref.	
Second remission	1.18 (0.80–1.74)	0.4	1.43 (1.05–1.94)	0.023	1.31 (1.03–1.66)	0.029	1.34 (1.05–1.71)	0.020
MFC MRD status before HCT								
MRD-negative	Ref.		Ref.		Ref.		Ref.	
MRD-positive	1.29 (0.84–1.97)	0.2	4.31 (3.31–5.62)	<0.001	2.95 (2.37–3.67)	<0.001	2.39 (1.91–2.99)	<0.001
Recovered peripheral blood counts before HCT	0.76 (0.55–1.05)	0.10	1.17 (0.89–1.53)	0.3	1.00 (0.81–1.22)	<0.9	0.91 (0.74–1.13)	0.4
HCT-CI category								
Low	Ref.		Ref.		Ref.		Ref.	
Intermediate	1.16 (0.79–1.68)	0.4	1.07 (0.80–1.43)	0.6	1.10 (0.87–1.38)	0.4	1.10 (0.86–1.40)	0.4
High	1.52 (1.03–2.23)	0.034	1.21 (0.88–1.65)	0.2	1.29 (1.01–1.65)	0.039	1.33 (1.04–1.72)	0.026
Stem cell source								
BM	Ref.		Ref.		Ref.		Ref.	
PBSC	1.34 (0.66–2.75)	0.4	0.64 (0.42–0.96)	0.033	0.78 (0.55–1.11)	0.2	0.85 (0.59–1.23)	0.4
Cord blood	2.24 (0.96–5.23)	0.062	0.46 (0.26–0.83)	0.009	0.77 (0.49–1.23)	0.3	0.97 (0.60–1.58)	>0.9
HLA matching								
Identical related donor	Ref.		Ref.		Ref.		Ref.	
Matched unrelated donor	0.94 (0.63–1.41)	0.8	0.89 (0.66–1.20)	0.4	0.90 (0.71–1.15)	0.4	0.93 (0.72–1.19)	0.5
1–2 allele mismatch	1.80 (1.11–2.93)	0.018	0.95 (0.62–1.47)	0.8	1.28 (0.93–1.76)	0.13	1.48 (1.06–2.05)	0.021
Haplo-identical	1.70 (0.68–4.24)	0.3	1.29 (0.72–2.32)	0.4	1.41 (0.86–2.30)	0.2	1.56 (0.92–2.65)	0.10
Conditioning regimen intensity								
MAC	Ref.		Ref.		Ref.		Ref.	
Non-MAC	1.51 (1.03–2.22)	0.033	1.71 (1.27–2.30)	<0.001	1.66 (1.31–2.09)	<0.001	1.48 (1.16–1.89)	0.002

Abbreviations: BM, bone marrow; ELN, European LeukemiaNet; HCT, hematopoietic cell transplantation; HCT-CI, HCT comorbidity index; HLA, human leukocyte antigen; MAC, myeloablative conditioning; MFC, multiparameter flow cytometry; MRC, U.K. Medical Research Council; MRD, measurable residual disease; PBSC, peripheral blood stem cells; WBC, white blood cell count.

**Table 4 cancers-15-00352-t004:** Univariable regression models of patients with AHD (*n* = 126).

	Non-Relapse Mortality		Relapse		RFS		OS	
	HR (95% CI)	*p*	HR (95% CI)	*p*	HR (95% CI)	*p*	HR (95% CI)	*p*
Age at transplantation, years	1.00 (0.97–1.03)	0.8	1.02 (0.99–1.04)	0.2	1.01 (0.99–1.03)	0.3	1.00 (0.98–1.02)	0.8
Female gender	0.57 (0.26–1.23)	0.2	0.80 (0.46–1.41)	0.4	0.71 (0.45–1.12)	0.14	0.66 (0.41–1.07)	0.089
WBC count at diagnosis, G/L	1.00 (0.99–1.01)	0.6	1.00 (0.99–1.01)	0.8	1.00 (0.99–1.01)	>0.9	1.00 (0.99–1.01)	0.9
Cytogenetic risk (MRC)								
Favorable								
Intermediate	Ref.		Ref.		Ref.		Ref.	
Adverse	0.51 (0.20–1.32)	0.2	1.80 (1.01–3.20)	0.046	1.18 (0.73–1.89)	0.5	1.01 (0.61–1.66)	>0.9
Time from last remission to HCT, days	1.00 (0.99–1.01)	>0.9	1.00 (1.00–1.01)	0.3	1.00 (1.00–1.01)	0.3	1.00 (1.00–1.00)	0.7
Disease status at HCT								
First remission	Ref.		Ref.		Ref.		Ref.	
Second remission	1.12 (0.39–3.22)	0.8	1.71 (0.86–3.42)	0.13	1.49 (0.84–2.65)	0.2	1.23 (0.68–2.24)	0.5
MFC MRD status before HCT								
MRD-negative	Ref.		Ref.		Ref.		Ref.	
MRD-positive	1.08 (0.52–2.25)	0.8	2.21 (1.28–3.81)	0.005	1.70 (1.10–2.61)	0.016	1.75 (1.12–2.72)	0.014
Recovered peripheral blood counts before HCT	0.91 (0.46–1.82)	0.8	1.26 (0.72–2.20)	0.4	1.11 (0.72–1.71)	0.6	1.10 (0.70–1.71)	0.7
HCT-CI category								
Low	Ref.		Ref.		Ref.		Ref.	
Intermediate	1.33 (0.57–3.11)	0.5	1.06 (0.50–2.22)	0.9	1.17 (0.67–2.04)	0.6	1.51 (0.83–2.75)	0.2
High	1.85 (0.76–4.49)	0.2	2.51 (1.27–4.94)	0.008	2.26 (1.32–3.86)	0.003	2.58 (1.45–4.58)	0.001
Stem cell source								
BM	Ref.		Ref.		Ref.		Ref.	
PBSC	0.21 (0.06–0.71)	0.012	1.54 (0.21–11.2)	0.7	0.54 (0.20–1.48)	0.2	0.38 (0.14–1.06)	0.066
Cord blood	0.52 (0.11–2.34)	0.4	2.68 (0.32–22.3)	0.4	1.07 (0.33–3.41)	>0.9	0.70 (0.21–2.29)	0.6
HLA matching								
Identical related donor	Ref.		Ref.		Ref.		Ref.	
Matched unrelated donor	1.12 (0.47–2.68)	0.8	1.70 (0.78–3.72)	0.2	1.43 (0.80–2.54)	0.2	1.14 (0.64–2.05)	0.7
1–2 allele mismatch	1.28 (0.42–3.95)	0.7	1.57 (0.57–4.33)	0.4	1.42 (0.67–3.02)	0.4	1.36 (0.63–2.93)	0.4
Haplo-identical	6.03 (0.69–52.5)	0.10	5.39 (1.11–26.1)	0.036	5.35 (1.51–19)	0.009	4.33 (1.24–15.1)	0.022
Cord blood	2.56 (0.75–8.74)	0.13	2.73 (0.94–7.90)	0.064	2.61 (1.17–5.81)	0.019	2.05 (0.90–4.68)	0.086
Conditioning regimen intensity								
MAC	Ref.		Ref.		Ref.		Ref.	
Non-MAC	0.88 (0.44–1.76)	0.7	2.11 (1.18–3.77)	0.012	1.48 (0.96–2.29)	0.075	1.06 (0.68–1.65)	0.8

Abbreviations: BM, bone marrow; ELN, European LeukemiaNet; HCT, hematopoietic cell transplantation; HCT-CI, HCT comorbidity index; HLA, human leukocyte antigen; MAC, myeloablative conditioning; MFC, multiparameter flow cytometry; MRC, U.K. Medical Research Council; MRD, measurable residual disease; PBSC, peripheral blood stem cells; WBC, white blood cell count.

## Data Availability

The datasets analyzed during the current study are available from the corresponding author upon reasonable request.

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
