# Peer review of "Association between Prior Cytotoxic Therapy, Antecedent Hematologic Disorder, and Outcome after Allogeneic Hematopoietic Cell Transplantation in Adult Acute Myeloid Leukemia"

_cancers, 2023, doi:10.3390/cancers15020352_

Round 1

Reviewer 1 Report

Corentin Orvain et al. investigated a large cohort of AML patients to examine the association between clinical history and AML outcomes in the context of allogeneic hematopoietic cell transplantation, found that AML after AHD but not therapy-related AML was associated with post-HCT outcomes in our cohort. The data are abundant, and the conclusions may benefit readers in AML treatment area, the reviewer only has minor issues listed below:

1. Please clarify which clinical center/centers are the patients from? Since this a large cohort.

2. The labels in Figures are too small.

Author Response

1) Please clarify which clinical center/centers are the patients from? Since this a large cohort.

All patients were treated at one institution (Fred Hutchinson Cancer Center/University of Washington/Seattle Cancer Care Alliance) in Seattle, USA. This is now explicitly stated in the Methods section, page 2: “…with or without peripheral blood count recovery between 5/2006 and 5/2021 at a single institution (Fred Hutchinson Cancer Center/University of Washington/Seattle Cancer Care Alliance, Seattle, WA, USA).

2) The labels in Figures are too small.

As suggested by this reviewer, we increased the size of labels for all figures.

Reviewer 2 Report

Very well written important study.

Well designed and comprehensively analyzed with clear discussion and conclusions.

I have no suggestions for improvement 

Author Response

We are glad that this reviewer found our work interesting, and we wish to thank him/her for critically reading our manuscript.

Reviewer 3 Report

Thank you for the opportunity to review the article "Association between prior cytotoxic therapy, antecedent hematologic disorder, and outcome after allogeneic hematopoietic cell transplantation in adult acute myeloid leukemia" by Orvain et al.

I commend the authors on their effort in this paper, that reports on 979 patients with AML (739 de novo, 125 with AHD and 115 therapy-related) undergoing first allo-HCT in 1st or 2nd remission. The main finding of this extensive work is that AHD has little impact on HCT outcomes, except for an increased NRM, mostly after MAC.

These results, clearly reported despite the bulk of data, may be highly relevant for the clinicians.

I suggest to further investigate and/or speculate on the role of age (significantly higher in the AHD cohort) on HCT "toxicity". It sounds a bit strange to me that, despite older age (and possibly higher burden of comorbidities), HCT-CI in the AHD group was not higher than in the de novo group. Can the authors confirm / discuss? Have the authors analyzed HCT NRM according to age at transplant (e.g., <60 or >60) in the AHD cohort? 

Author Response

I suggest to further investigate and/or speculate on the role of age (significantly higher in the AHD cohort) on HCT "toxicity". It sounds a bit strange to me that, despite older age (and possibly higher burden of comorbidities), HCT-CI in the AHD group was not higher than in the de novo group. Can the authors confirm / discuss? Have the authors analyzed HCT NRM according to age at transplant (e.g., <60 or >60) in the AHD cohort? 

We confirm that patient with AHD were older but had a similar number of comorbidities, as evaluated by the HCT-CI. We may speculate that physicians in our institution may be more reluctant proceeding to allogeneic HCT in patients with AHD if they are older and have many comorbidities. This was added to the Discussion section, page 14: Despite being older, patients with AHD had a similar number of comorbidities, as assessed by the HCT-CI. This may represent a bias toward selecting fitter subsets of patients for allografting.Although age was not associated with non-relapse mortality in patients with AHD, we specifically explored the association between age and post-HCT outcomes. These results were added to the Results section, page 8: “As patients with AHD were significantly older, we specifically explored the association between age and post-HCT outcomes, dichotomizing patients into those younger vs. older than 60 years old. While the NRM for younger and older patients was similar (at three years: 22% [10-34%] vs. 19% [7-31%], P=0.99), as was OS (at three years: 55% [42-71%] vs. 38% [29-52%], P=0.38), relapse was non-significantly higher (at three years: 26% [14-39%] vs. 48% [37-60%], P=0.08) and RFS was non-significantly lower (at three years: 52% [39-68%] vs. 33% [24-45%], P=0.07) in younger patients.